# Optimizing Strategies for the Urban Work Zone with Time Window Constraints

**Yao Yu [1],\*, Jinxian Weng [1] and Wanying Zhu [2]**

[1] College of Transport and Communications, Shanghai Maritime University, Shanghai 201306, China
[2] Vehicle Manufacturing Engineering, SAIC General Motors Co. Ltd., Shanghai 201206, China
\* Correspondence: yaoyu@shmtu.edu.cn

**Abstract:** Work zones that move with road maintenance tasks are enclosing and have caused severe traffic jams and the significant decline of road capacity. This paper proposes an intelligent-based multi-objects road maintenance optimization strategy based on a practical origin–destination (OD) matrix and complicated work schedules over a real urban road network. It focuses on the optimization of multi short-term maintenance tasks and the minimization of average travel delay for vehicles passing through. By taking the driving characteristic into account, static and dynamic variable speed limit strategies provide access to ensure safety on the working road network. Through this view, the problem was formulated as a mixed multi-object nonlinear program (MNLP) model with respect to the time window of the related sub-maintenance task. By using actual OD distribution matrix data, a series of microscopic simulated cases were conducted to test the model's validity. Moreover, sensitive analyses of types of parameters (e.g., traffic safety threshold, traffic flow and working efficiency) with an optimal solution were discussed considering five different scenarios.

**Keywords:** work zone; time window; traffic delay; traffic safety

---

## 1. Introduction

As a result of rapid economization and motorization, urban roads and their ancillary facilities have to be maintained regularly in order to suit the increment of traffic demand. Regular maintenance tasks have been proved as an effective method for keeping road service at a good level. As mentioned below, maintenance projects are referred to as work zones. It should be pointed out that work zones often necessitate the closure of one or more traffic lanes to protect the safety of the workers, resulting in serve traffic congestion during peak hours [1–3]. Consequently, it is critically important to optimize and implement a reasonable work strategy for the urban work zone.

In general, the work zone operation strategy has a great impact on maintenance costs as well as road user costs, such as travel time. Currently, most of the previous studies have focused on how to minimize costs from the contractor's perspective. However, the high efficiency of maintenance means that more lanes need to be enclosed along the work zone, which results in reducing traveling speed and road segment capacity. Accordingly, the probability of traffic accident risk will increase [4]. Instead, from the views of the transport management authority, this study employs three categorized factors, namely, the work zone maintenance time window, network road safety and the work team workload, and innovatively introduces speed limit control strategies [5], aiming to decrease traffic flow speed variation, thereby improving traffic safety for the whole road segment. Therefore, an intelligent-based multi-objects road maintenance optimization strategy is proposed using actual origin–destination (OD) datasets over a real urban road network to improve the efficiency of work maintenance while decreasing the cost of travel and emission.

## 2. Relevant Studies

In recent decades, most of the studies used optimization methods and various models have been proposed. The main decision objective of optimal maintenance work zone is concentrated on minimizing road accident risk as well as travel delay and total maintenance cost. Some of those models aimed to the optimize single work zone operational strategy without using a speed limit control method and project start time. For example, Schonfeld et al. [6] established an unconstrained optimization model to minimize the average cost by determining the optimum length of dual-lane highways. More recently, Chien et al. [7] extended their proposed model to a four-lane highway work zone with one lane closure. Similarly, Chen et al. [8] proposed a novel method on four-lane bidirectional highways by setting two-lane pavements as alternative routes. In addition, Bo et al. [9] used the temporary shoulder in short-term maintenance work under different traffic conditions and found that a shoulder could improve the efficiency of the work zone.

Time-varying demands have been taken into account in the work zone optimization problem in recent studies. For example, [10] used neural networks to optimize work zone lengths and starting time for short-term road maintenance on highways when processed by an average daily traffic volumes (ADT) database. Their objective was to minimize the average work zone cost. However, it should be pointed out that the delay cost estimation formula used in their work should be remedied when the queue vanished within a time interval. To improve the accuracy of the estimation formula, Meng et al. [11] designed a model in terms of a time window constraint to examine the impact of unequal sub-work-zone lengths with an actual roadside tree trimming project. Huang et al. [12] applied a trial-and-error method to find the optimal work zone length and the best work starting time to minimize the total cost of maintenance and delays. Their results showed that the optimal starting time was not sensitive to the flow speed. However, apparently, the increment of work zone average passing speed significantly reduces the total cost (travel time and emission) under congested condition.

Traffic safety is the key premise of short-term maintenance efficiency. Considering traffic safety and construction and maintenance issues, Ullman et al. [13] conducted a study for determining key work zone safety and mobility performance measures. Shakouri et al. [14] investigated the effect of changing work zone configurations and traffic density on performance in terms of variables and subjective workload. Actually, due to the difficulty of the quantization for traffic safety, another factor, vehicle speed, was used to replace it. Kanf et al. [15] proposed a speed control strategy named the time-of-day speed limit control. The proposed method was limited by the prediction accuracy, though it could overtake the difficulty in setting the optimal real-time speed limit. It was a new attempt to improve the driving safety by speed control. Xu et al. [16] found a critical parameter of safety from classical models which could describe the behavior of vehicles while passing through work zone.

In addition, it is noted that the microscopic simulated method is a feasible approach to find the optimal solution under an actual control strategy with variable speed limit advice. For example, Chen and Scriba et al. [17,18] both used micro-simulation to find the best strategy of work zone without considering the actual working load.

## 3. Problem Description

The problem can be described as follows: (i) typical urban road work zones contain *N* short-term maintenance projects and *M* work teams; (ii) each team can only be assigned one task at any time. In addition, all short-term projects are required to be completed within a predetermined time window. Moreover, in order to reduce traffic delay and enhance traffic safety, an effective variable speed limit strategy is implemented in each work zone. We formulated the problem as a mixed multi-object nonlinear program (MNLP) model with respect to the time window of the related sub-maintenance task. Due to the large dimension of the proposed problem and the complexity of the nonlinear model, solving the MNLP directly was challenging. To simplify the problem, we defined the starting time and number of work teams as an interactive pair of parameters and designed a genetic algorithm to resolve the model accordingly. It is noted that all data can be acquired under intelligent road circumstances.

Summarizing the above, the $i$th work zone maintenance project consists of three main parameters: (i) $x_i$, the starting time of the $i$th work zone; (ii) the work team which is to undertake the $i$th work zone maintenance task and (iii) the speed limit control strategy set at beginning of the $i$th work zone.

The vector $X$ denotes the set of starting times of all the work zones, from $x_1$ to $x_n$. Similarly, vectors $Y$ and $Z$ illustrate the sets of work teams and speed limit control strategies, respectively, which are associated with the work zone maintenance projects. Therefore, the maintenance schedule $S$ can be draw as

$$S = \begin{cases} X : (x_1, x_2 \cdots x_n) \\ Y : (y_1, y_2 \cdots y_n) \\ Z : (z_1, z_2 \cdots z_n) \end{cases} \tag{1}$$

## 4. Speed Limit Control Strategy

In this paper, we propose three speed limit control strategies which are able to ensure the traffic safety in work zones, including one conventional speed limit control strategy and two variable speed limit control strategies.

As a kind of well-used and conventional speed limit control strategy, fixed speed limit marks are adopted in our simulated fields. With reference to "Road traffic signs and markings-Part 4 work zone (GB5768.4-2009)" and "Traffic safety signs of road operation (GA182-1998)" in China [19,20], we determined the laying schemes of the fixed speed limit signs in all the work zones, such as the positions and the boundary ranges of the speed limit values, etc.

Additionally, to test the effectiveness, the above-mentioned variable speed limit (VSL) control strategies are applied in different control scheme of algorithms, which name the VSL-I and the VSL-II respectively. First, algorithm VSL-I is on the basis of research in [21], which considered the fluctuation of approaching traffic demand. The objective of the VSL-I is to minimize the queue length of the work zone, consisting of two modules. As shown in Figure 1, the first module (Module 1) function computes the initial speed of each VSL control stretch. Then, the second module (Module 2) is responsible for updating the control speed in terms of the difference between the detected flow speed and the target control speed over fixed time steps.

The traffic stream is specified as homogenous. The essential principle for Module 1 is that the traffic flow rate begins at the initial road segment and discharging at the lane-closing segment is consistent. The variation parameter of the mean density of subzone1 $D_1$ (upstream of work zone) should be estimated according to the subzone traffic flow $F_1$ and work zone $F_0$ during each time interval $\Delta t$ as Equation (2) shows. Then the display speed $V_i$ can be estimated by the traffic flow and density.

$$\begin{cases} D_1 = D_1(k-1) + \frac{F_0(k) - F_1(k)}{L} \Delta t \\ V_i = \frac{F_i}{D_i}, i = 1, 2 \ldots n \end{cases} \tag{2}$$

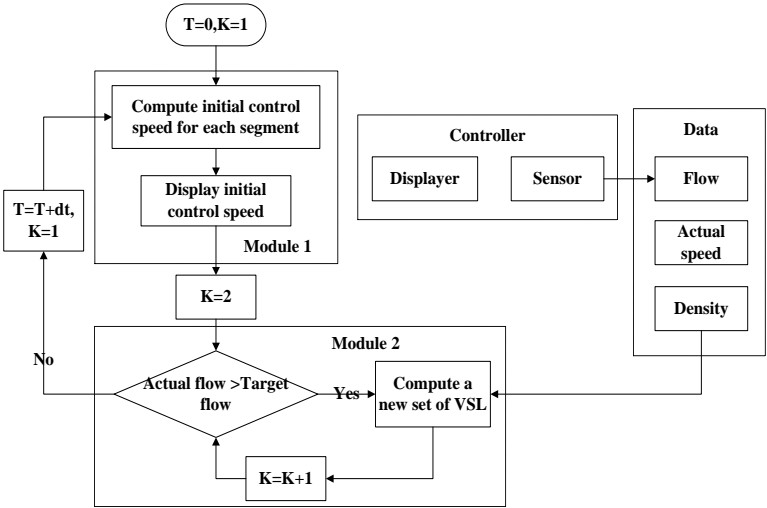

**Figure 1.** A flow chart of the VSL algorithm.

Due to the various characteristics of driving behaviors, the compliance rate needs to be taken into account. Therefore, the aim of Module 2 is to improve the acceptability of the VSL by acquiring the speed difference between the actual speed and the target speed in order to update the displayed speeds accordingly. The algorithm uses identical parameters to maximize the flow over the work zone, which constrains by the target volume and safety requirement. The step-by-step procedures are shown in Figure 2.

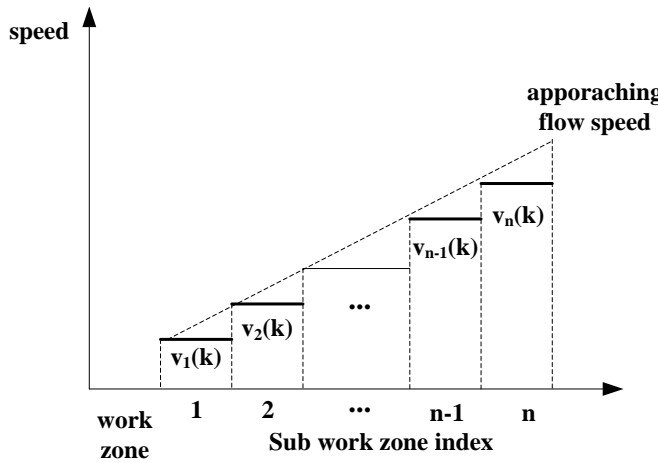

**Figure 2.** Ideal relationship between control speed and displayed speed.

ALINEA, a local feedback ramp-metering strategy control strategy can be considered as one type of VSL control algorithm, proposed by Papageorgiou [4], that has proved to be a remarkably simple, efficient and easily implemented method in ramp metering. The logic in ALINEA is to alleviate the gap between the actual detected density and desired density, as shown in Figure 3.

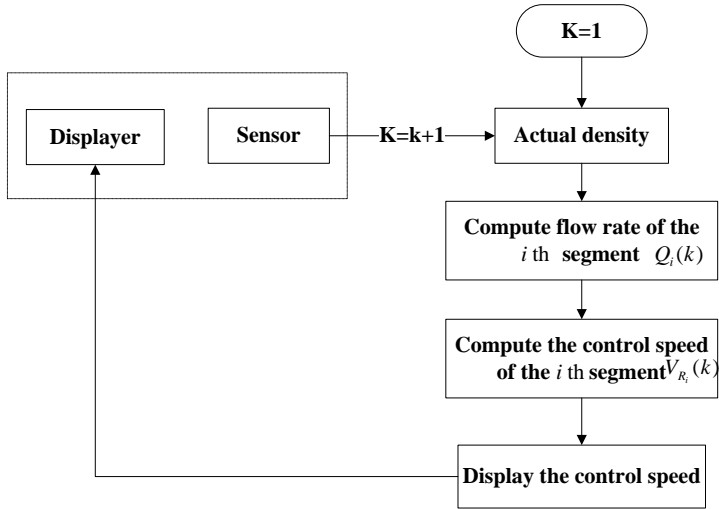

**Figure 3.** Flow chart of ALINEA algorithm.

In Figure 3, the actual density is calculated by the detected real-time traffic parameter upstream. Consequently, the downstream flow and an update speed can be estimated. It is noted that ALINEA includes a dynamic iterate feedback procedure and the accuracy of data acquisition has a great impact on the speed *V(k)* on target segment.

## 5. Key Constraints Description

### 5.1. Traffic Safety Quantitation

Speed drop has a significant correlation with safety. Here, we consider safety as a constraint parameter quantified by the average speed variance of vehicles passing through the target work zone. Safety $f_s$ is formulated as

$$f_s = \sum_{i=1}^{n} (v_i - \overline{v})^2 / N \tag{3}$$

Figure 4 shows a simplified variable speed detection schematic diagram, which depends on the advised speed instruction with respect to the preceding vehicle in a continuous time step.

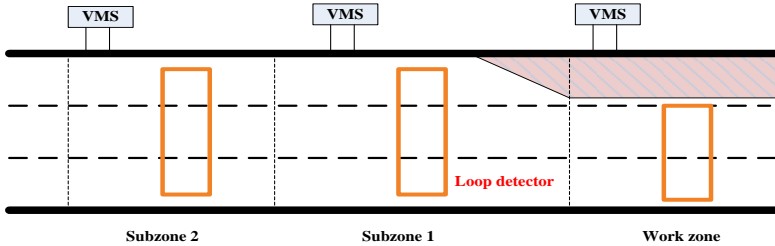

**Figure 4.** Simplified variable speed detection schematic diagram.

### 5.2. Traffic Delay Estimation

To minimize the average travel delay of traffic flow, we assumed that traffic delay is the function of flow and determined by different maintenance tasks. The specific value of travel time can be obtained by the following equation:

$$D(X, Y, Z) = \frac{1}{K}\sum_{k=1}^{K} t_k^1 - \sum_{k=1}^{K} t_k^0 \tag{4}$$

where $D(X, Y, Z)$ represents the average travel delay of the network under the maintenance schedule $S = (X, Y, Z)$. $K$ is the total number of vehicles in the network, and $t_k^0$ denotes the individual travel time of the $k$th vehicle in the network. $t_k^1$ exhibits the travel time of the $k$th vehicle when work zones begin and maintenance activities are implemented in reference to the schedule $S = (X, Y, Z)$.

## 6. Optimization Model

As aforementioned, the objective minimization model is defined as

$$\min D(X, Y, Z) = \frac{1}{K} \sum_{k=1}^{K} t_k^1 - \sum_{k=1}^{K} t_k^0 \tag{5}$$

The proposed object function is required to satisfy the following constraints:

(i) Time window constraint

$$u_i^0 \leq u_i \leq u_i^1 - a_i, \quad i = 1, 2 \cdots n \tag{6}$$

where $u_i^0$ and $u_i^1$ are integers denoting the earliest starting time and the finish time for the $i$th work zone project, respectively. $u_i$ denotes the starting time of the $i$th work zone and $a_i$ is scheduled maintenance time.

(ii) Traffic safety constraint

$$f(S) \leq f_s \tag{7}$$

Equation (7) illustrates that each work zone optimal strategy should not exceed the lowest requirement threshold of safety.

(iii) Workload constraint

$$0 < \sum_{i=1}^{n} h_{ij} \leq l_j, \quad i = 1, 2 \cdots n, \, j = 1, 2 \cdots m \tag{8}$$

$$\sum_{j=1}^{m} h_{ij} = 1, \quad i = 1, 2 \cdots n, \, j = 1, 2 \cdots m \tag{9}$$

where $h_{ij}$ is the binary variable that denotes the matching value of work zones and work teams. $l_j$ presents the maximum number of maintenance projects that can be assigned to the $j$th work team. It is noted that each work team can only be assigned one maintain task during a time window.

In addition, we define some assumptions as:

(i) Due to the short-term work zone projects, drivers may not know the changes of traffic environment in advance. Their trip routes were not taken into account in the current road conditions.

(ii) The proportion of heavy vehicles type is constant.

(iii) The moving time of the work team for consecutive maintain tasks is not taken into consideration.

## 7. Numerical Results

The proposed model in Section 6 can be regarded as a mixed multi-object nonlinear program (MNLP) problem. Due to the large dimension of the proposed problem and the complexity of the nonlinear model, we adopted an improved GA (Genetic Algorithm) to resolve with respect to the time window pair of starting time and number work teams. The test network involves six work zones and three work teams in Beixiaguan district, in Beijing. Six maintenance work zones are distributed on different road segments with details shown in Table 1.

**Table 1.** Work zones attributes.

| Work Zone | Maintenance Time (Hours) | Lane Closure | Time Window (O'clock) |
|-----------|--------------------------|--------------|------------------------|
| 1 | 2 | Right and Middle | [5, 19] |
| 2 | 2 | Right | [7, 18] |
| 3 | 3 | Left and Middle | [5, 20] |
| 4 | 3 | Right | [8, 20] |
| 5 | 1 | Right and Middle | [6, 19] |
| 6 | 1 | Left | [5, 15] |

Similarly, the work team attributes are shown in Table 2 with columns for the maximum work hours and time window. Besides, there are three alternative speed limit control strategies for work teams to undergo maintenance projects. The final scheduling contains the following results:

(i)　work team responsible for maintenance project in each work zone;
(ii)　start time of each maintenance project;
(iii)　speed limit control strategy, which is adopted in each work zone maintenance project.

**Table 2.** Work team attributes.

| Work Team | Maximum Load (Hours) | Work Time Window (O'clock) |
|-----------|----------------------|-----------------------------|
| 1 | 2 | [4, 24] |
| 2 | 2 | [4, 24] |
| 3 | 3 | [4, 24] |

## 7.1. Simulated Parameters

We picked up a partial road network in Beixiaguan district, in Beijing, with 74 nodes, 160 links and eight signalized intersections. The OD trip distribution is updated per hour based on the actual traffic flow between 18:00 to 19:00 as an OD based matrix (ground matrix) as Table 3 shows. Furthermore, all signalized phases adopt an actuated signal timing strategy. The cycle time ranges from 60 to 210 s according to the overall traffic demand variation.

**Table 3.** Origin–destination (OD)-based matrix (time from 18:00 to 19:00).

| O | D | | | | | | | | |
|------|--------|--------|--------|--------|--------|--------|--------|--------|-------|
| | Zone 1 | Zone 2 | Zone 3 | Zone 4 | Zone 5 | Zone 6 | Zone 7 | Zone 8 | Total |
| Zone 1 | 0 | 60 | 52 | 56 | 25 | 45 | 104 | 93 | 435 |
| Zone 2 | 145 | 0 | 115 | 124 | 44 | 64 | 74 | 160 | 726 |
| Zone 3 | 27 | 42 | 0 | 103 | 137 | 112 | 94 | 49 | 564 |
| Zone 4 | 140 | 129 | 118 | 0 | 104 | 120 | 78 | 148 | 837 |
| Zone 5 | 112 | 81 | 174 | 184 | 0 | 47 | 113 | 88 | 799 |
| Zone 6 | 80 | 33 | 109 | 81 | 55 | 0 | 103 | 29 | 490 |
| Zone 7 | 57 | 25 | 93 | 85 | 95 | 102 | 0 | 64 | 521 |
| Zone 8 | 172 | 122 | 118 | 109 | 36 | 60 | 78 | 0 | 695 |
| Total | 733 | 492 | 779 | 742 | 496 | 550 | 644 | 631 | 5067 |

All-or-nothing assignment rule was applied. Specifically, assuming that all maintenance activities are short-term tasks and drivers wouldn't change their route even if lane closure in downstream. Table 4 summarizes the specification of the parameter assignment employed in this study.

**Table 4.** Parameter list.

| Parameter | Description | Input Value | Type |
|-----------|-------------|-------------|------|
| $f_s$ | Traffic safety threshold | 25 | model parameter |
| R | GA population size | 4 | |
| $p_c$ | GA crossover probability | 0.8 | |
| $p_m$ | GA mutation probability | 0.25 | algorithm parameters |
| G | Number of generations to stop at | 100 | |
| h | Target headway in simulation | 1.5 | |
| $T_r$ | Mean reaction time in simulation | 1 | simulation parameters |
| g | Minimum gap in simulation | 1.5 | |

### 7.2. Case Analysis

The optimization procedure was processed by the mentioned GA algorithm. The convergence process is shown in Figure 5. Each point in the following figures presents the numbers of generation iteration in the GA algorithm. The calculated progress of the average travel delay presents a staged decreasing trend and becomes stable from the 35th generation. Table 5 displays the variation of global optimal solutions as generation iteration increment. And the convergence results showed the final maintenance schedule was obtained at the 35th generation. In detail, the global optimal solution was not found until 35 iteration times as the average travel delay improved from 790s to 550s compared to the initial scheme.

Obviously, in Table 5, six work zone maintenance programs were assigned to three work teams. Thus, team 1 was assigned only one task. However, team 3 had to finish three tasks and was the "busiest" team. The conventional speed limit control strategy was applied in two-thirds of the work zones to reduce the potential risk effectively. It is noted that the speeds of most the vehicles that approached the work zone are lower than the advised speed. That phenomenon accounts for the speed limit control strategy that has little effect on speed variances in the urban road segment.

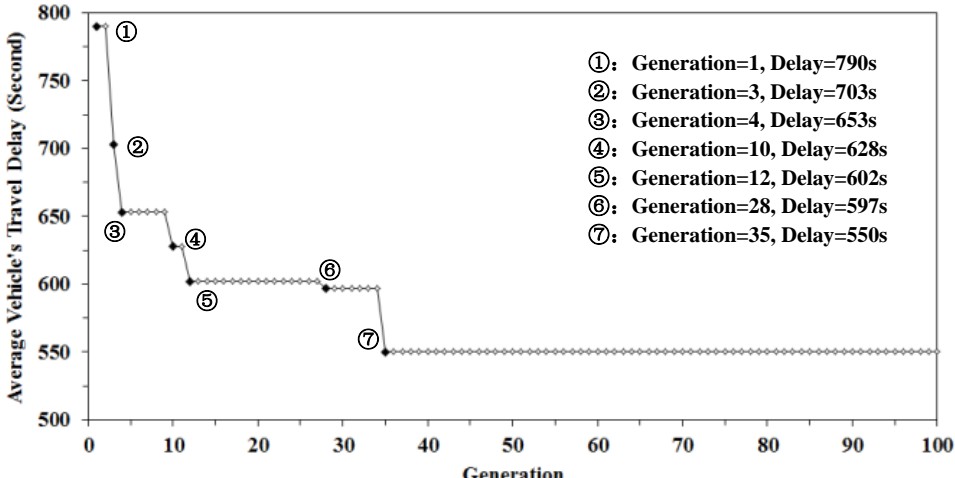

**Figure 5.** Convergence of GA search.

**Table 5.** Best schedules in each generation.

| Generation | Work Zone | Work Team | Maintenance Time (O'clock) | Speed Limit Control Strategy | Delay |
|---|---|---|---|---|---|
| 1–2 | 1 | 2 | [15, 17] | 0 | 790 |
| | 2 | 3 | [10, 12] | 2 | |
| | 3 | 1 | [7, 10] | 0 | |
| | 4 | 2 | [17, 20] | 2 | |
| | 5 | 1 | [17, 18] | 1 | |
| | 6 | 3 | [8, 9] | 1 | |
| 3 | 1 | 2 | [15, 17] | 0 | 703 |
| | 2 | 3 | [11, 13] | 2 | |
| | 3 | 1 | [7, 10] | 0 | |
| | 4 | 2 | [17, 20] | 0 | |
| | 5 | 3 | [17, 18] | 1 | |
| | 6 | 3 | [8, 9] | 1 | |
| 4–9 | 1 | 2 | [15, 17] | 0 | 653 |
| | 2 | 2 | [11, 13] | 2 | |
| | 3 | 1 | [7, 10] | 0 | |
| | 4 | 3 | [17, 20] | 0 | |
| | 5 | 3 | [15, 16] | 1 | |
| | 6 | 3 | [9, 10] | 1 | |
| 10–11 | 1 | 2 | [15, 17] | 0 | 628 |
| | 2 | 2 | [11, 13] | 2 | |
| | 3 | 1 | [7, 10] | 0 | |
| | 4 | 3 | [17, 20] | 0 | |
| | 5 | 3 | [15, 16] | 1 | |
| | 6 | 3 | [10, 11] | 1 | |
| 12–27 | 1 | 2 | [15, 17] | 0 | 602 |
| | 2 | 2 | [11, 13] | 2 | |
| | 3 | 1 | [8, 11] | 0 | |
| | 4 | 3 | [17, 20] | 0 | |
| | 5 | 3 | [15, 16] | 1 | |
| | 6 | 3 | [8, 9] | 0 | |
| 28–34 | 1 | 2 | [14, 16] | 0 | 597 |
| | 2 | 3 | [11, 13] | 2 | |
| | 3 | 2 | [10, 12] | 1 | |
| | 4 | 1 | [17, 20] | 0 | |
| | 5 | 3 | [14, 17] | 1 | |
| | 6 | 3 | [8, 9] | 0 | |
| 35–100 | 1 | 2 | [14, 16] | 0 | 550 |
| | 2 | 3 | [11, 13] | 2 | |
| | 3 | 1 | [9, 12] | 0 | |
| | 4 | 2 | [17, 20] | 0 | |
| | 5 | 3 | [15, 16] | 1 | |
| | 6 | 3 | [8, 9] | 0 | |

*7.3. Parameters Analyzed*

Moreover, to explore the relationships between sets of given factors (i.e., $f_s$, $Q$ and efficiency $E$) and average travel delay $D$, several scenarios were designed with VSL. $f_s^B$, $Q^B$ and $E^B$ are three basic constants, which represent the initial traffic safety, network traffic volume and work team efficiency, respectively. In the following text, a total of five alternative scenarios were undertaken with the following specifications in order to capture the correlation between the safety and travel delay under the urban road network circumstance.

Scenario 1: $f_s^1 = 0.5f_s^B$, $Q^1 = Q^B$, $E^1 = E^B$
Scenario 2: $f_s^2 = 0.75f_s^B$, $Q^2 = Q^B$, $E^2 = E^B$
Scenario 3: $f_s^3 = f_s^B$, $Q^3 = Q^B$, $E^3 = E^B$
Scenario 4: $f_s^4 = 1.25f_s^B$, $Q^4 = Q^B$, $E^4 = E^B$
Scenario 5: $f_s^5 = 1.5f_s^B$, $Q^5 = Q^B$, $E^5 = E^B$

Figure 6 demonstrates the GA search convergence and average travel delay in different scenarios. Obviously, the largest value of travel delay is 724 s (in scenario 1), which is approximately 1.5 times the smallest value (485 s in scenario 5). Of note is the larger safety threshold value set and the smaller average travel delay of the work zone and vice versa. On the other hand, the curves can also be explained by the fact that the speed fluctuation is dependent on the safety toleration of upstream vehicles in reference to formula 3. Therefore, a significant result can be concluded that the traffic safety threshold value has a positive impact on the speed variance of vehicles passing through the designated work zone. Moreover, the VSL strategy is one of the key factors that influences the variance of speed, which should not be neglected.

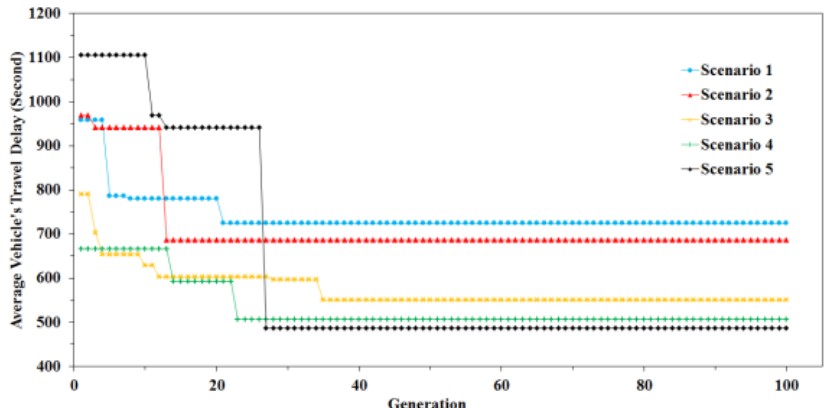

**Figure 6.** GA search convergence and final average travel delay under different scenarios.

Similarly, the parameter of traffic flow also has a potential impact on the optimal results. Thus, another five scenarios were used to acquire the characteristic between *Q* and *D*.

Scenario 1: $f_s^1 = f_s^B$, $Q^1 = 0.5Q^B$, $E^1 = E^B$
Scenario 2: $f_s^2 = f_s^B$, $Q^2 = 0.75Q^B$, $E^2 = E^B$
Scenario 3: $f_s^3 = f_s^B$, $Q^3 = Q^B$, $E^3 = E^B$
Scenario 4: $f_s^4 = f_s^B$, $Q^4 = 1.25Q^B$, $E^4 = E^B$
Scenario 5: $f_s^5 = f_s^B$, $Q^5 = 1.5Q^B$, $E^5 = E^B$

The above calculated numerical results are showed in Figure 7, which indicates that there is an existing non-linear positive correlation between traffic delay and traffic flow. Specifically, in Scenario 1, traffic flow is only half the initial value; however, it shows a significant decrease from 550 (ground value in scenario 3) to 48 s for travel delay as well as the same significant decreasing trend in scenario 2 than ground truth. That means traffic flow plays an important role in the downtown district of the urban road network because of its huge travel demand and prosperous economics. Therefore, a better optimal result can be exhibited only if the assumed traffic volume is lower than the saturation condition. On the other hand, the delay has a slight increase (for example, from 790 s in scenario 4 to 800 s in scenario 5) over saturation condition.

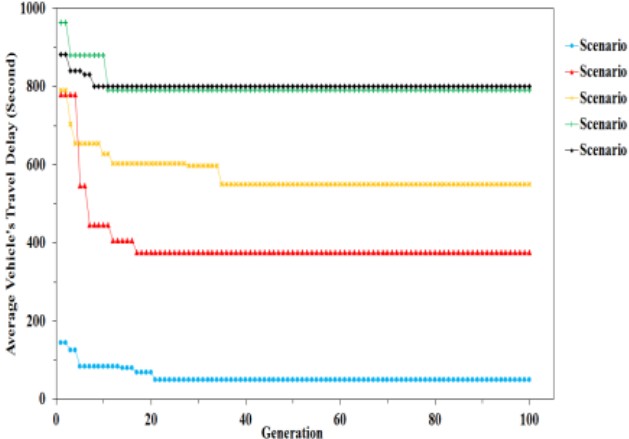

**Figure 7.** Sensitive analysis of traffic flow.

Finally, working efficiency is an important factor of work zone maintenance activities as it can affect the maintenance time directly. For example, according to Table 5, it takes two hours for a work team to complete the Work Zone 1 maintenance job with normal working efficiency. Furthermore, the time will be expanded to four hours if the working efficiency reduces to fifty percent. Based on that, we set five scenarios as

Scenario 1: $f_s^5 = f_s^B$, $Q^5 = Q^B$, $E^5 = 1.5E^B$
Scenario 2: $f_s^4 = f_s^B$, $Q^4 = Q^B$, $E^4 = 1.25E^B$
Scenario 3: $f_s^3 = f_s^B$, $Q^3 = Q^B$, $E^3 = E^B$
Scenario 4: $f_s^2 = f_s^B$, $Q^2 = Q^B$, $E^2 = 0.75E^B$
Scenario 5: $f_s^1 = f_s^B$, $Q^1 = Q^B$, $E^1 = 0.5E^B$

It is can be seen from Figure 8 that a higher working efficiency makes a contribution to shorten the work zone maintenance time and cost. In addition, the curves show a downward trend with the increment of the working efficiency. A markable reduction in travel delay occurs when the control parameter efficiency increases by 50%, as shown in Scenario 1 in Figure 8. A highly efficient operation ensures the rapid completion of the work zone and minimizes the negative impact of lane closure on normal traffic in urban road networks. Conversely, travel delay will increase by 55% and 29% compared to the ground truth (scenario 3) corresponding to the efficiency drops by 50% (scenario 5) and 75% (scenario 4), respectively.

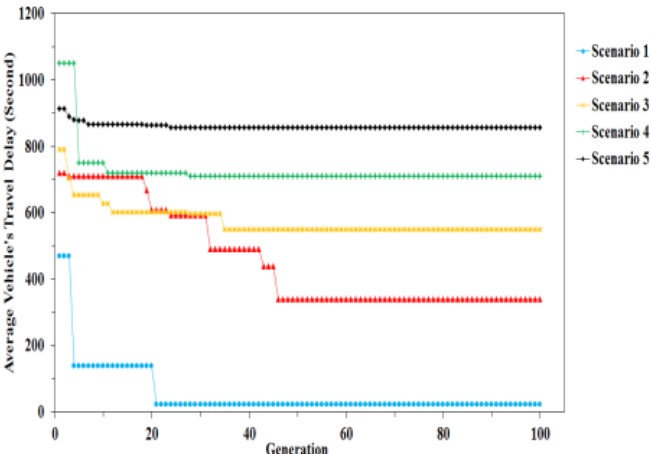

**Figure 8.** Sensitive analysis of work team efficiency.

## 8. Conclusions

In this paper, a mixed multi-object nonlinear program model is proposed with respect to a time window of short-term maintenance tasks over a real urban road network. The constraints of traffic safety and maintenance efficiency, etc., are taken into account to optimize the accuracy of the maintenance schedule and minimize travel delay while vehicles pass through work zone segments. By employing a genetic algorithm, we obtained specific optimal results including (i) six short-term tasks assignment for three work teams; (ii) the starting time for each maintenance task and (iii) the mode of the speed limit control strategy adopted for different work zones. Then, we evaluated the performance of the proposed model using a simulation with actual OD data, which proved that there is over a 30% improvement on travel delay compared to the initial (actual) maintenance plan. Furthermore, to capture the impact on object function travel delay, three key parameters, namely, safety, traffic flow and efficiency, are discussed respectively. The simulated results indicate that the above constraints have significant impact on final optimal results which appear to have inversely proportional to the threshold value of safety and traffic flow. Specifically, the delay of traffic flow has a tiny increase, approximately less than 2% under the over saturation traffic condition. However, working efficiency is positive correlated with maintenance cost, which increases by 55% and 29% than the ground bases while efficiency drops by 50% and 75% respectively. Moreover, the VSL strategy plays a crucial role in the speed variation of traffic flow. It helps drivers decelerate before accessing the work zone. For this, the boundary value of the speed limit is discussed by considering the parameter of safety.

**Author Contributions:** Y.Y. and J.W. conceived of the presented idea and developed the model and simulations. W.Z. assisted with OD data and results analysis. All authors discussed the results and commented on the manuscript.

**Funding:** This study is supported by the "National Natural Science Foundation of China" (grant No. 61603247 and 71871137) and the Shanghai Education Development Foundation and the Shanghai Municipal Education Commission (grant No. 16SG41).

**Conflicts of Interest:** The authors declare no conflict of interest in their research.

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
