# Peer review of "Optimizing Strategies for the Urban Work Zone with Time Window Constraints"

_sustainability, doi:10.3390/su11154218_

Round 1
Reviewer 1 Report
interesting and good quality paper
For a multi-objective problem (working time, road safety, transit time), it was necessary to integrate the other elements of sustainable development, in particular the measurement of greenhouse gas emissions which is fundamentally tied to the duration of transit and to the speed vihecules.
can the choice of areas to be treated in the paper be justified?
Are nearby commercial activities, including delivery and pick-up operations, taken into account?
the same question for the nature of the vehicles. the difference between a light vehicle, a tricycle, a bicycle, a goods transport vehicle and a heavy weight is very important. is this parameter taken into account?
Author Response
Thanks for your observations and comments:
Point 1: Can the choice of areas to be treated in the paper be justified?
Response 1: The answer is yes. We adopted a complicated test in reality city road network which is flexible and accessible. However, necessities of OD data and other attributions of work zones are the premise to acquire if we choice different areas.
Point 2: Are nearby commercial activities, including delivery and pick-up operations, taken into account?
Response 2: The proposed example road-network includes one university and several office buildings located at downtown district with heavy traffic in daily. In specific environment, we didn’t figure commercial activities like delivery and pick-up hot-points etc. which can induce more pedestrian and trucks mixed traffic flow. The reason is heavy and motor vehicle types are restricted in downtown district in Beijing. But this is a good view to be considered in our further research in urban road network.
Point 3: The same question for the nature of the vehicles. the difference between a light vehicle, a tricycle, a bicycle, a goods transport vehicle and a heavy weight is very important. is this parameter taken into account?
Response 2: Various types of vehicles have great impact on efficiency of work zones. In PARAMICS simulated process, we take different vehicles types which classified by DWT into account without considering tricycle and bicycle since their uncertain behavior and difficulty to measure. Here, the accuracy of proposed model will not influenced if the signal optimal is not taken into account. And now we focus on how to display a novel method to integrate work zone missions and road traffic. We will consider other traffic modes in further research combined with signal optimal in road network.
Please see the attachment about the above response.

Reviewer 2 Report
The article deals with an interesting topic and fits for Journal’s aim. The issue discussed in this study is original with reference to contemporary scientific discussion the urban transport. The study highlights intelligent-based multi objects road maintenance optimization strategy based on practical origin-destination matrix and complicated work schedules over reality urban road network..
In my opinion, the study should not be accepted for publication in its present version since it presents a number of caveats. In a reviser version of the papier, the author should carefully address the following points.
1. Introduction should inform about the aim of manuscript.
2. The paradigms of smart city is key-concepts of the paper. However, there is no way to detect which is the author’s conceptual approach to there paradigm.
3. The literature review should contain references concerning the methodology used in the study. Moreover, I would suggest the author put in evidence analogies and differences of related studies with respect to the methodology which implements in this study.
4. The results should be clearly present (for instance in points).
5. I would suggest the author make the reader aware of the possible reasons of the results.
Author Response
Thanks for your observations and comments:
Point 1: Introduction should inform about the aim of manuscript.
Response 1: We update aim of this paper in introduction at line 39 to 42 as “……Therefore, an intelligent-based multi objects road maintenance optimization strategy is proposed using actual OD datasets over reality urban road network to improve the efficiency of work maintenance while decreasing the cost of travel and emission.” with red marked.
Point 2: The paradigms of smart city is key-concepts of the paper. However, there is no way to detect which is the author’s conceptual approach to there paradigm.
Response 2: The real-time traffic data is necessity for our proposed method. So here we made an assumption that all data we need can be acquired in urban road network. In this way, there is accessible to optimal result step by step, and variable speed strategies can effective working. Therefore, we revised data assumption annotation in PROBLEM DESCRIBTION part in line 92 to 93 as “……It is noted that all data can be acquired under intelligent road circumstance……”
Point 3: The literature review should contain references concerning the methodology used in the study. Moreover, I would suggest the author put in evidence analogies and differences of related studies with respect to the methodology which implements in this study.
Response 3: Up to now, most studies of work zone problems adopted optimal methodology as references show in this paper. The significant differences of these studies in literature review are optimization goals as well as signal objective or multi-objectives which displayed in line 46/49/54/56/58/62/etc. So we make a supplement of methodology illustration at the initial of literature review as “……most of studies using optimization method and various models have been proposed……”.
Point 4: The results should be clearly present (for instance in points).
Response 4: Each point in result figures represents numbers of generation iteration in GA algorithm. We add this illustration in part 7.2 as “…Each point in following figures presents numbers of generation iteration in GA algorithm…..” to make the reader better understand the point meaning in Figures.
Point 5: I would suggest the author make the reader aware of the possible reasons of the results.
Response 5: It is important to make readers clearly know the reasons of optimal results. So we add demonstrations as “……Obviously, the largest value of travel delay is 724 seconds (in scenario 1) as approximate 1.5 times as the smallest one (485s in scenario 5). It is point that, the larger safety threshold value set, the smaller average travel delay for work zone maintenance schedule and vice versa. In other hand, the curves also can be explained that the speed fluctuation is depending on the safety toleration of upstream vehicles in refers to formula 3……” in Part 7.3;
Also add “……Specifically, in Scenario 1, traffic flow is only half than the initial value however it shows a prominent decrease slash from 550s (ground value in scenario 3) to 48s for travel delay as well as same significant decreasing trend in scenario 2 than ground truth. That means traffic flow plays an important role in downtown district of urban road-network because of its huge travel demand and prosperous economics. Therefore, a better optimal result can be exhibited only if the assumed traffic volume less than saturation condition. On the other hand, the delay has a slightly increase (for example, from 790s in scenario 4 to 800s in scenario 5) under over saturation condition……” above the Fig.7.
And “……And the curves show downward trend with the increment of the working efficiency. A markable reduction of travel delay occurs when the control parameter efficiency increase 50%, as shown in Scenario 1 in Fig.8. High efficient operation ensures rapid completion of work zone and minimizes the negative impact of lane closure on normal traffic in urban……” before CONCLUSION.
